# Assessing roost disturbance of straw-coloured fruit bats (*Eidolon helvum*) through tri-axial acceleration

Tânia Domingues Costa[1]☉*, Carlos D. Santos[2,3]☉*, Ana Rainho[1], Michael Abedi-Lartey[3,4], Jakob Fahr[3,5], Martin Wikelski[3,4], Dina K. N. Dechmann[3,4]

**1** cE3c —Centre for Ecology, Evolution and Environmental Changes, Departamento de Biologia Animal, Faculdade de Ciências, Universidade de Lisboa, Lisboa, Portugal, **2** Núcleo de Teoria e Pesquisa do Comportamento, Universidade Federal do Pará, Rua Augusto Correa 1, Guamá, Belém, Brazil, **3** Department of Migration, Max Planck Institute of Animal Behavior, Radolfzell, Germany, **4** Department of Biology, University of Konstanz, Konstanz, Germany, **5** Braunschweig University of Technology, Zoological Institute, Braunschweig, Germany

☉ These authors contributed equally to this work.
* taniadominguescosta@gmail.com (TDC); cdsantos@ab.mpg.de (CDS)

**Data Availability Statement:** The full tracking dataset is available at the Movebank Data Repository (https://doi.org/10.5441/001/1.k8n02jn8).

## Abstract

The disturbance of wildlife by humans is a worldwide phenomenon that contributes to the loss of biodiversity. It can impact animals' behaviour and physiology, and this can lead to changes in species distribution and richness. Wildlife disturbance has mostly been assessed through direct observation. However, advances in bio-logging provide a new range of sensors that may allow measuring disturbance of animals with high precision and remotely, and reducing the effects of human observers. We used tri-axial accelerometers to identify daytime flights of roosting straw-coloured fruit bats (*Eidolon helvum*), which were used as a proxy for roost disturbance. This bat species roosts on trees in large numbers (often reaching hundreds of thousands of animals), making them highly vulnerable to disturbance. We captured and tagged 46 straw-coloured fruit bats with dataloggers, containing a global positioning system (GPS) and an accelerometer, in five roosts in Ghana, Burkina Faso and Zambia. Daytime roost flights were identified from accelerometer signatures and modelled against our activity in the roosts during the days of trapping, as a predictor of roost disturbance, and natural stressors (solar irradiance, precipitation and wind speed). We found that daytime roost flight probability increased during days of trapping and with increasing solar irradiance (which may reflect the search for shade to prevent overheating). Our results validate the use of accelerometers to measure roost disturbance of straw-coloured fruit bats and suggest that these devices may be very useful in conservation monitoring programs for large fruit bat species.

## Introduction

Wildlife disturbance from human activities is a global threat contributing to the loss of biodiversity [1]. This threat has spread to remote natural regions, and is a common problem in protected areas for wildlife conservation [2, 3]. Disturbance may lead to changes in animals'

**Funding:** This study was supported by the Max Planck Institute of Animal Behavior, the Max Planck Society, and field work in Zambia 2014 was supported through funds to the Institute of Novel and Emerging Infectious Diseases (Prof. Dr. Martin H. Groschup, Friedrich-Loeffler-Institute, Greifswald, Germany) from the Federal Foreign Office of Germany (ref # ZMVI6-FKZ2513AA0374). The funders had no role in study design, data collection and analysis, decision to publish, or preparation of the manuscript.

**Competing interests:** The authors have declared that no competing interests exist.

activity patterns [4, 5], energy expenditure [6, 7], physiological parameters [7, 8], foraging behaviour [9, 10], breeding success [11] and roosting behaviour [12, 13]. Ultimately, it can drive changes in species distributions and richness [1, 14]. However, assessing the impacts of disturbance on wildlife is a challenging task, as impacts vary across species and contexts [14]. Sophisticated methods have been employed, such as measuring changes in stress hormones, cardiac response and immunocompetence [15]. However, the direct observation of changes in behaviour is still the most prevalent approach described in the literature [16].

The use of automated methods in behavioural studies of wildlife disturbance, such as infrared motion detectors [17], radio-telemetry [7, 10, 18] and global positioning system (GPS) tracking [19], has become more common in recent years. These methods provide large amounts of accurate data and reduce the influence of the observer on the behaviour of the target animals [20], although biologging methods may cause some disturbance to the animals [21]. Further innovative applications are expected in the near future from advances in animal tracking technology that make available a range of new sensors to measure behavioural parameters [22, 23]. Tri-axial accelerometers, in particular, are present in most modern tracking devices, allowing precise measurement of animals' body motion, from which different behaviours can be discriminated [23]. However, to our knowledge, these sensors have never been used to measure animal disturbance.

The straw-coloured fruit bat (*Eidolon helvum*) is a large Old World fruit bat species (Pteropodidae) that occurs across sub-Saharan Africa [24]. It feeds upon a large variety of fruits and flowers, often moving tens of kilometres between the roost and the foraging areas on a daily basis [25]. These features contribute to making this species a keystone seed disperser in Africa, and critical for maintaining vegetation dynamics in fragmented tropical forests [26]. Despite its high ecological relevance, populations of straw-coloured fruit bat are declining across its range, with hunting probably being the main cause [27–29]. This species roosts on trees in large numbers, reaching hundreds of thousands of individuals [13], with an estimated five to ten million bats in the largest known colony at Kasanka National Park (Zambia) [30]. Very often roosts are located in urban areas [13, 31], making them especially vulnerable to interactions with humans [28, 29, 32].

New and exact information of disturbance levels in roosts of this species is crucial to inform conservation actions and protection regulations. This study aimed to examine the potential of tri-axial accelerometery to monitor disturbance of roosting bats. For that purpose, we tagged 46 straw-coloured fruit bats with GPS-accelerometer dataloggers in five roosts in Ghana, Burkina Faso and Zambia. We identified daytime flights in roosts from accelerometer signatures and use them as a proxy of roost disturbance. Flight is an extremely energy-consuming activity for bats, demanding up to 34 times the basal metabolic rate [33], thus we expect them to avoid flying during the roosting period. The occurrence of daytime roost flights was then modelled against our activity in the roosts during the days of trapping, as a predictor of roost disturbance, and natural stressors that may influence flight probability during the roosting period. We predicted that: (1) our presence in the roosts will lead to higher daytime flight probability; (2) bats will be more prone to fly in days of high solar irradiance to find shaded perches and avoid heat stress [34, 35]; (3) bats will fly less during daytime when it rains, as rain increases flight energy costs [36]; (4) bats will fly less at higher wind speeds, which might increase flight energy costs [37].

## Materials and methods

### Ethics statement

Fieldwork, including bat handling and tagging, was approved by Ghana Wildlife Division of the Forestry Commission (permit FCWD/GH-01), Zambia Wildlife Authority (permits

ZAWA 421902 and ZAWA 547649), and the Director of the Parc Urbain Bangr-Weoogo (Mr Moustapha Sarr).

## Study areas

Straw-coloured fruit bats were captured and tagged with dataloggers in five roosts at four different areas across continental Africa: Accra and Kibi in Ghana, Ouagadougou in Burkina Faso, and Kasanka National Park in Zambia (Fig 1).

In Accra, bats were captured in a roost near the city centre, in the area of the 37 Military Hospital (5.586˚N, 0.185˚W). Accra is one of the largest cities in West Africa with almost two million people. The area around the city still holds remnants of coastal savanna forest, but is dominated by introduced tree species. The colony varies in size across seasons: it peaks during the dry season, reaching 100,000–250,000 individuals, and only a few thousand individuals are present during the wet season [25, 31].

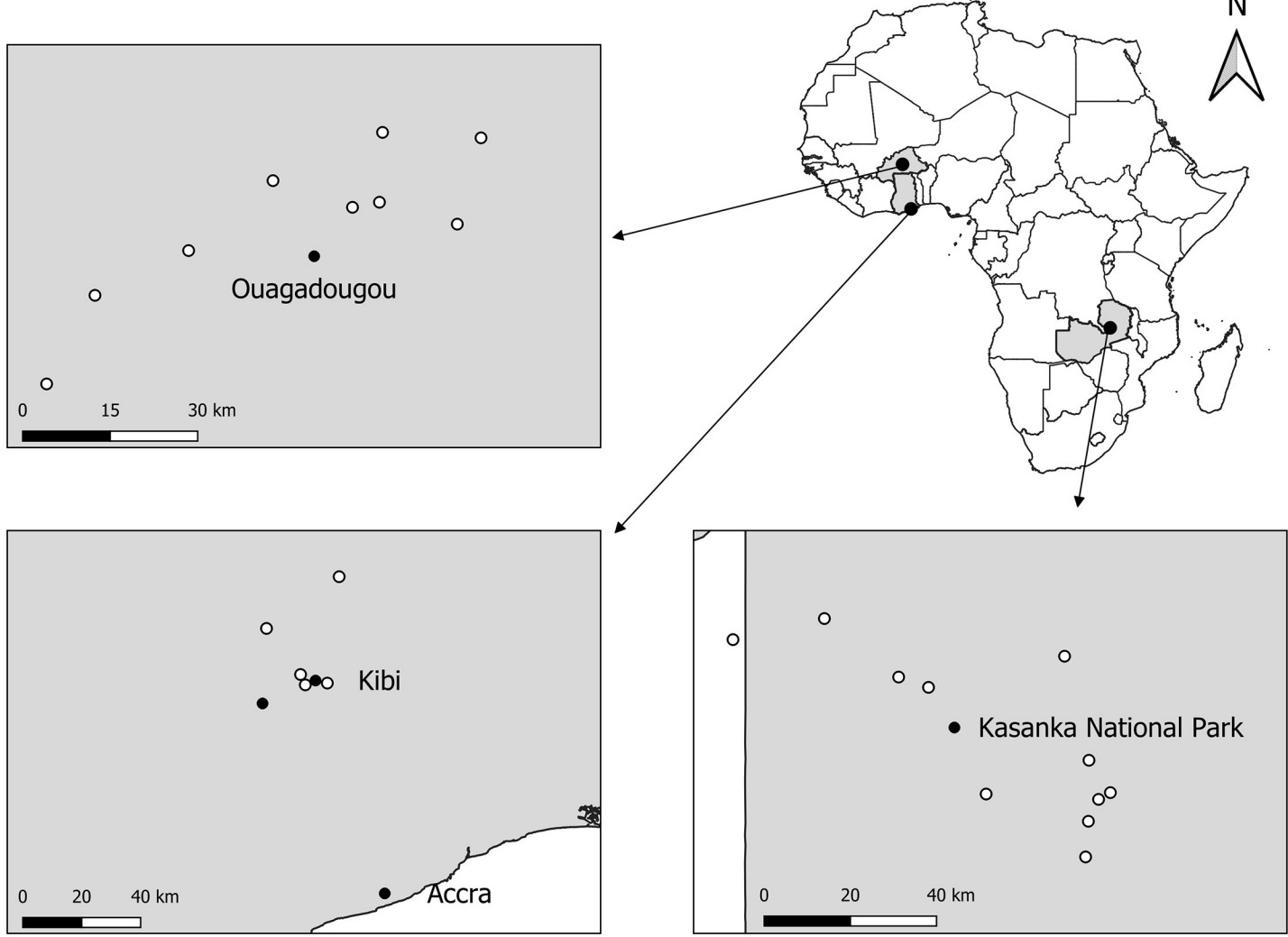

**Fig 1. Location of straw-coloured fruit bat roosts studied at Burkina Faso (top left), Ghana (bottom left), and Zambia (bottom right).** Black dots represent roosts where bats were captured and tagged with tracking devices and white dots represent other roosts used by tagged bats. Background map, provided by GADM (https://gadm.org), is licensed for academic use.

In Kibi, bats were captured in two neighbouring roosts (ca. 19 km apart), one in Old Tafo (6.235˚N, 0.394˚W) and the other at Kibi Palace (6.165˚N, 0.555˚W). Kibi is a rural area with ca. 168,000 people, covered by moist semi-deciduous forests, farmlands and degraded forests [31]. The colonies peak during the dry season with a total of 40,000–50,000 individuals and the numbers decline during the wet season down to a low of a few hundred individuals [31].

In Ouagadougou bats were captured in a roost located in an urban park near the city centre (Parc Urbain Bangr Weogo, 12.398˚N, 1.489˚W). This city has ca. 1.5 million people and is located in the savanna biome [38]. During monthly counts undertaken in 2013 and 2014, this colony peaked during the wet season with 70,000–125,000 individuals while the roost was vacated during the dry season [38].

In Kasanka National Park, bats were captured in the largest colony known for this species, with an estimated peak of ten million individuals [30]. The park covers an area of 420 km$^2$, dominated by Miombo forests [30], and the roost site is located in a patch of Mushitu swamp forest near the Fibwe Campsite (12.587˚S, 30.242˚E). In this roost, bats are present only from October to December (wet season) [30]. The region has a low population density (14 people per km$^2$) with about 85% of the population living in rural areas.

## Bat capture and tracking

We tagged bats between 2009 and 2014 during different years for each study area and both wet and dry seasons in Kibi and Accra (Table 1). We netted bats in the morning (3:00 to 06:00), as they returned from foraging. We weighted the captured bats, determined their age and sex and measured the length of the forearm. Dataloggers (20–24 g e-obs GmbH, Munich, Germany) were fitted only on large individuals (239 to 321 g) to minimize effects of the extra load on their behaviour. Most tagged bats were adult males (43 individuals), but we also tagged two young males and one adult female (S1 Table of S1 File). We attached dataloggers with glue (Sauer Hautkleber, Manfred Sauer GmbH) to the back of the bat (for 12 individuals) or with a neck collar made of goat leather and closed with degradable suture thread (for 34 individuals, S1 Table of S1 File, [25, 31, 39]). The weight of the datalogger and collar (when used) ranged from 6.9 to 10.5% of the bats' body mass (mean: 8.5%). Dataloggers recorded GPS locations only during the night (18:00 to 6:00, at least every 30 min), but tri-axial acceleration was recorded around-the-clock in bursts of 13 or 14s per min at 20 or 18.74 Hz depending on the logger generation (S1 Table of S1 File). Data were retrieved using a base station connected to a directional high-gain antenna. Further details on field procedures and the tracking devices can be found in earlier studies that used data from the same bats [25, 26, 31, 37, 40].

**Table 1. Summary of the periods of data collection and the number of bats tracked in each study area.**

| Area | Year | Season | Dates of collection | Number of bats |
|---|---|---|---|---|
| Accra | 2009 | Wet | 26/08–31/08 | 6 |
| | 2011 | Dry | 02/02–09/02 | 4 |
| Kibi Palace | 2011 | Wet | 27/08–31/08 | 2 |
| Kibi Old Tafo | 2012 | Wet | 28/08–16/09 | 4 |
| | 2013 | Dry | 25/01–01/02 | 3 |
| | 2013 | Wet | 20/09–24/09 | 1 |
| Ouagadougou | 2013 | Wet | 19/08–31/08 | 4 |
| | 2014 | Wet | 17/06–24/06 | 6 |
| Kasanka | 2013 | Wet | 04/12–11/12 | 3 |
| | 2014 | Wet | 29/11–08/11 | 13 |

## Roost disturbance estimation

We used flight events during daytime (7:00 to 17:00) as a proxy of roost disturbance. Daytime roost flights were detected from acceleration readings with high variation in heave compared with surge and sway (Fig 2). During flight, the body of the bat shows regular vertical oscillation of high amplitude and lower variation on the lateral and longitudinal planes (Fig 2, [23]). Specifically, acceleration bursts were classified as "flying" if they matched the following criteria:

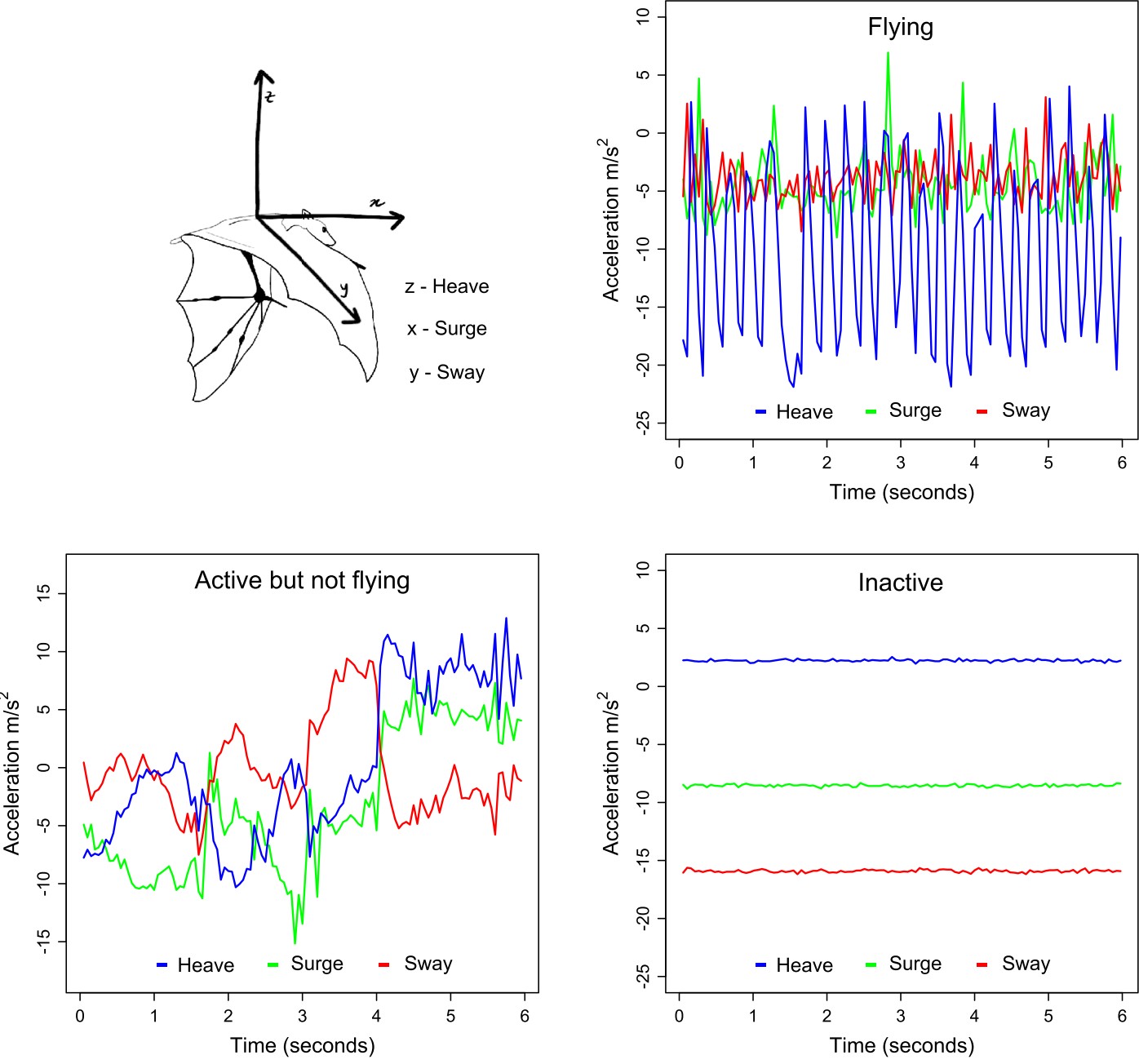

**Fig 2. Representation of tri-axial accelerometer attached to a bat and respective axes (z–heave, x–surge, y–sway) and acceleration signatures of different behaviours.** Illustration by Sara Gomes based on a photograph of Mark Carwardine.

(1) the mean of heave amplitudes calculated for each second was higher than 10 m/s$^2$, this value being higher than the corresponding values of surge and sway; (2) the mean of heave values was lower than those of surge and sway. Criterion (1) ensured that the datalogger had oscillation of high amplitude in the vertical axis, but not as much in the other axes (see Fig 2 top right plot). Criterion (2) ensured that the datalogger was upright in case the bat was flying. An upside-down placement of the datalogger would increase heave values by 19.6 m/s$^2$ (i.e. 2 g units), making them higher than those of surge and sway (see Fig 2 bottom right plot as an example). We also considered as "flying" the acceleration bursts that matched the above criteria only at their beginning or ending sections, which we interpreted as landing or departure events. Cases of dubious oscillation patterns, suggesting that the bat was flying with the datalogger wrongly positioned were excluded from analyses. We applied classification criteria to the data with an R [41] routine but we validated visually all acceleration signatures classified as "flying" and those with high oscillation of any axes that were classified as "not flying". We also applied this classification to a subsample of data that included GPS locations (i.e. collected after 18:00) to confirm that bats with acceleration signatures classified as "flying" were indeed moving. Although flight behaviour was classified every minute, we aggregated classifications each day because of the small number of bursts classified as "flying". Thus, the variable used in the analysis was binary, representing the occurrence of daytime roost flights for each animal in each day of tracking. We excluded data from bats that moved more than 500 m during the day (identified by comparing the morning and evening GPS fixes), as we could not accurately define which roost they spent the day. We also excluded data from the first day of tracking for each animal, as we expected its behaviour to be affected by the recent capture and handling.

## Predictors of roost disturbance and natural stressors

We tested the effects of our presence in the roosts during the days of trapping and a set of environmental variables on the probably of bats to fly during the day. Although there was no time overlap between the data used for analyses and our visits to the roosts (i.e. we left one hour before data were collected for analyses), we assumed that our presence had a lasting disturbance effect. Bats were captured with mist nets set at the level of the canopy, which disturbed animals that were roosting in the nearby trees, and we expected their escape flights to spread throughout the roost affecting bats tagged in previous trapping sessions. We also assumed that solar irradiance, precipitation and wind speed could potentially influence daytime flight behaviour of bats in the roosts based on earlier studies [34–37]. These variables were obtained from open access weather databases (http://www.sasscalweathernet.org for Zambia roosts and https://globalweather.tamu.edu for all the others) with a temporal resolution of one day. For roosts located close together, the data was retrieved from the same weather stations. This was the case of smaller roosts located around the main roosts where bats were captured (white dots in Fig 1), and also for the two main roosts located in Kibi (Fig 1). We did not include temperature as predictor in our models because daily mean values were not available for all study areas.

## Modelling procedures

We evaluated the effects of our presence in the roosts during trapping days and weather variables on the probability of bats to undertake diurnal flights at the roost with two Generalized Linear Mixed Models (GLMMs). The first included the full data set, the second excluded the data from trapping days. For both models, the occurrence of daytime roost flights was included as the dependent variable, and individual ID and roost ID were included as random intercept factors. The first model included trapping day, solar irradiance, precipitation and wind speed

as fixed effects. Trapping day was binary (trapping days vs regular days) and the remaining variables were continuous. The second model used all variables but trapping day as fixed effects. Models were fitted with the function glmer of the R-package lme4 [42]. Marginal and conditional R squared were calculated with the function r.squaredGLMM of the MuMIn R-package [43]. Temporal autocorrelations of model residuals were generally low, not requiring further corrections (S1 Fig of S1 File).

### Data accessibility

The full tracking dataset is available at the Movebank Data Repository [44].

### Results

We retrieved daytime flight data from 46 individuals during one to seven days, providing 167 observations (S1 Table of S1 File). Among these, 129 were recorded at roosts where captures took place and 38 at other roosts. Daytime roost flights were identified from acceleration data in 24 cases.

The model including the full dataset showed significant effects of our presence in the roosts during trapping days and solar irradiation on the probability of bats to exhibit diurnal flights in the roost (Table 2). During the days of trapping bats were more likely to fly in the roost than on regular days (Fig 3, Table 2). Daytime roost flight probability also increased with solar irradiance, with a more pronounced effect when solar irradiance exceeded 20 MJ/m$^2$ (Fig 3, Table 2). The remaining variables did not affect the diurnal flight of bats at the roost (Table 2).

When excluding the data recorded during the days of trapping, only solar irradiation showed a significant effect on daytime roost flight probability, with an increasing pattern similar to the first model (Table 2). The dataset used in this model contained 131 observations, 36 less than the first model.

### Discussion

Our study reports and validates the use of tri-axial accelerometry as a novel approach to monitor roost disturbance of large bats. Although animal tracking devices allow precise recording of animal behaviour and reduce the influence of human observers, they have rarely been used to monitor animal disturbance [but see 7, 10, 12, 18, 19]. Among sensors in tracking devices, tri-axial accelerators are particularly effective because they record behavioural data at high

**Table 2. Summary of binomial GLMMs testing the effects of environmental variables on the probability of straw-coloured fruit bats to fly at their roosts during the day.** The response variable was assigned as 1 for the days when the bats flew in the roost and 0 otherwise. The first model included days when we trapped bats with mist nets in the roosts, therefore we included trapping day as a binary model predictor (trapping days vs regular days). Both models included individual ID and roost ID as random intercept factors. Marginal and conditional R$^2$ were calculated with the function r.squaredGLMM of the MuMIn R-package [43]. Significant relationships are shown in bold and are plotted in Fig 2. Units of parameter range: Solar irradiance—MJ/m$^2$; Precipitation—mm/day; Wind speed—m/s.

| Model | Parameter | Range | Estimate | SE | Z | P-value | R$^2$ cond./marg |
|---|---|---|---|---|---|---|---|
| With trapping days | Intercept | - | -7.164 | 2.005 | -3.57 | >0.001 | 0.192/0.175 |
| | Trapping day | 0–1 | 1.634 | 0.543 | 3.01 | **0.003** | |
| | Solar irradiance | 2.5–30.7 | 0.236 | 0.089 | 2.66 | **0.008** | |
| | Precipitation | 0.0–87.0 | 0.044 | 0.041 | 1.08 | 0.280 | |
| | Wind speed | 0.5–4.0 | -0.335 | 0.368 | -0.91 | 0.363 | |
| Without trapping days | Intercept | - | -6.008 | 1.831 | -3.28 | 0.001 | 0.06/0.06 |
| | Solar irradiance | 2.5–30.7 | 0.175 | 0.089 | 1.96 | **0.050** | |
| | Precipitation | 0.0–87.0 | 0.034 | 0.042 | 0.82 | 0.410 | |
| | Wind speed | 0.5–4.0 | -0.144 | 0.429 | -0.34 | 0.738 | |

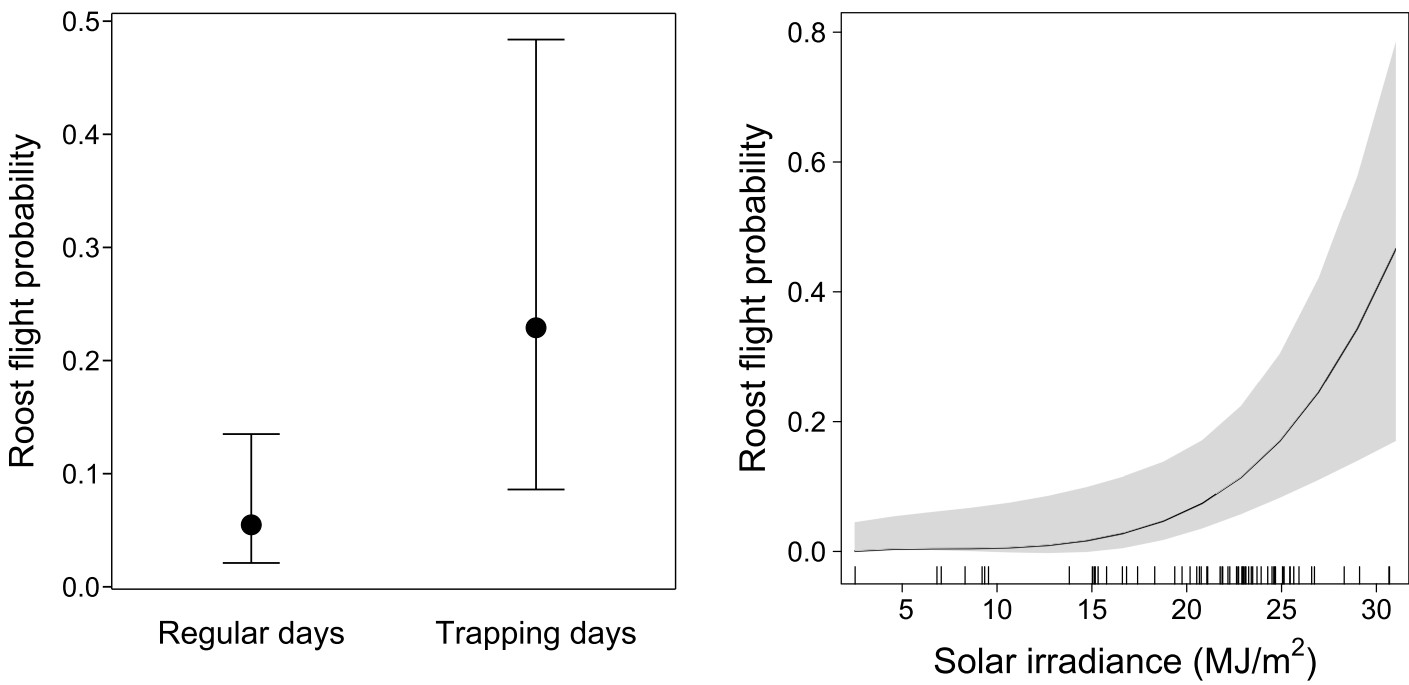

**Fig 3. Model partial effects of our presence in the roosts during trapping days and solar irradiation on the probability of bats to undertake diurnal flights at the roost.** The model is a binomial GLMM that also includes wind speed and precipitation as predictors, and individual ID and roost ID as random intercept factors (see Table 2). Error bars (left plot) and shading areas (middle and right plots) represent 95% confidence intervals.

frequencies and demand low battery power to operate. This contrasts with the recording of GPS fixes that require higher battery power, thus delivering coarser measurements of animal behaviour.

We showed that diurnal roost flights were more likely to happen during the days we trapped bats in the roosts (Fig 3). This result validates the use of tri-axial accelerometry to monitor bat roost disturbance by humans as our presence in the roosts during trapping days was an unequivocal cause of disturbance. Hunting may have a comparable effect in the roosts as hunters use firearms and nets to reach bats roosting high in the trees [13, 29]. Besides direct mortality, these methods cause an overall disturbance of the roost. Therefore, our method has high potential to monitor hunting in bat roosts, an activity that is assumed to cause population declines of straw-coloured fruit bats [27–29].

Among the remaining factors tested as model predictors, only solar irradiance showed a significant effect. Bats were more likely to fly within the roost during the day with increasing solar irradiance, particularly when this variable exceeded 20 MJ/m$^2$ (Fig 3). This may reflect bats seeking shade to prevent overheating, particularly on very hot days. This thermoregulatory behaviour has been described in Australian flying fox species (*Pteropus alecto* and *Pteropus poliocephalus*) when exposed to high temperatures [35]. We had also expected bats to fly less within the roost during days with precipitation, as rainfall increases flight energy costs [36]. However, events of significant precipitation were relatively rare in our sample, only in 7% of the sampling days it rained more than 10 mm. This likely prevented the identification of an effect of precipitation in our models. We also did not find an effect of wind on the probability of diurnal flights at the roost. This shows that, against our expectations, bats did not avoid

flying on windy days, at least within the range of wind speed observed during data collection (0.5 to 4 m/s).

The model that excluded data recorded during the trapping days showed a poorer fit than the previous (Table 2). In this model, solar irradiance had a significant relationship with day-time flight probability at the roost, which showed a similar pattern to that of the first model. The weakness of this model was expected given the considerable reduction in sample size (36 observations less than the original dataset).

We must emphasize that our models did not explore all factors that could contribute to the disturbance of roosting bats. Large bat colonies are likely to attract non-human predators [45], but we were unable to assess the significance of this potential disturbance factor. We also did not evaluate hunting as a factor of disturbance, although this is assumed to be one of the main causes of population declines in straw-coloured fruit bats [27–29]. The exclusion of these potentially important factors may have influenced the precision of our models (Table 2).

Although straw-coloured fruit bats are still relatively abundant across their distribution range, their role as a seed disperser may be seriously impacted by ongoing population declines [27, 29], with consequences for the maintenance and regeneration of tropical forests in Africa [26]. The fact that this species aggregates in large roosts often located in urban areas makes it particularly vulnerable to human pressures, thus the monitoring of detrimental factors in these roosts is of utmost importance. We believe that the method presented here can be an effective solution in conservation monitoring programs of large fruit bats, particularly to monitor unprotected roosts located in remote areas. It can also potentially be used for poacher detection, perhaps combined with motion capture sensors and other methods already being used [46]. Tracking devices have evolved considerably since our data collection. Most are now solar charged, extending their lifespan, and can send data remotely (by GSM or Satellite), which reduces fieldwork effort and prevents the loss of data. In addition, their costs have reduced considerably, making them accessible to conservation projects with relatively modest budgets. Thus, the method described here can certainly be implemented at better cost-efficiency in the current days and in the near future.

## Supporting information

**S1 File.**
(PDF)

## Acknowledgments

We thank Richard Suu-Ire for help with logistics and permits in Ghana, Lackson Chama for help with permits in Zambia, and Frank Willems, Sebastian Stockmaier and Natalie Weber for help in the field in Zambia. We thank Sarah Davidson and the Movebank team for data curation. We thank Sara Gomes for the bat illustration in Fig 2 and Mark Carwardine for providing the original photograph. We also thank João Paulo Silva, Hariprasath Ramesh and the three reviewers for comments on the manuscript.

## Author Contributions

**Conceptualization:** Tânia Domingues Costa, Carlos D. Santos, Ana Rainho, Martin Wikelski, Dina K. N. Dechmann.

**Data curation:** Carlos D. Santos, Michael Abedi-Lartey, Jakob Fahr.

**Formal analysis:** Tânia Domingues Costa, Carlos D. Santos, Ana Rainho.

**Funding acquisition:** Jakob Fahr, Martin Wikelski, Dina K. N. Dechmann.

**Investigation:** Tânia Domingues Costa, Carlos D. Santos, Ana Rainho, Michael Abedi-Lartey, Jakob Fahr, Martin Wikelski, Dina K. N. Dechmann.

**Methodology:** Tânia Domingues Costa, Carlos D. Santos, Ana Rainho, Michael Abedi-Lartey, Jakob Fahr, Dina K. N. Dechmann.

**Project administration:** Jakob Fahr, Dina K. N. Dechmann.

**Resources:** Dina K. N. Dechmann.

**Supervision:** Carlos D. Santos, Ana Rainho, Jakob Fahr, Dina K. N. Dechmann.

**Validation:** Tânia Domingues Costa, Carlos D. Santos, Ana Rainho.

**Visualization:** Tânia Domingues Costa, Carlos D. Santos, Ana Rainho.

**Writing – original draft:** Tânia Domingues Costa, Carlos D. Santos.

**Writing – review & editing:** Tânia Domingues Costa, Carlos D. Santos, Ana Rainho, Michael Abedi-Lartey, Jakob Fahr, Martin Wikelski, Dina K. N. Dechmann.

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
