## [Decision Letter · Decision Letter 0]

18 Aug 2020

PONE-D-20-15313

Assessing roost disturbance of straw-coloured fruit bats (Eidolon helvum) through tri-axial acceleration

PLOS ONE

Dear Dr. Santos,

Thank you for submitting your manuscript to PLOS ONE. After careful consideration, we feel that it has merit but does not fully meet PLOS ONE’s publication criteria as it currently stands. Therefore, we invite you to submit a revised version of the manuscript that addresses the points raised during the review process.

We look forward to receiving your revised manuscript.

Kind regards,

Christian Vincenot, Ph.D.

Academic Editor

PLOS ONE

Additional Editor Comments:

All the reviewers agree that the study is sound and interesting. Please address all the minor suggestions and also see particularly to the insightful comment by reviewers 2 and 3 on the statistical analysis.

Journal Requirements:

2. We note that Figure 1 in your submission contain map images which may be copyrighted. All PLOS content is published under the Creative Commons Attribution License (CC BY 4.0), which means that the manuscript, images, and Supporting Information files will be freely available online, and any third party is permitted to access, download, copy, distribute, and use these materials in any way, even commercially, with proper attribution. For these reasons, we cannot publish previously copyrighted maps or satellite images created using proprietary data, such as Google software (Google Maps, Street View, and Earth). For more information, see our copyright guidelines: http://journals.plos.org/plosone/s/licenses-and-copyright.

2.1.    You may seek permission from the original copyright holder of Figure 1 to publish the content specifically under the CC BY 4.0 license. 

2.2.    If you are unable to obtain permission from the original copyright holder to publish these figures under the CC BY 4.0 license or if the copyright holder’s requirements are incompatible with the CC BY 4.0 license, please either i) remove the figure or ii) supply a replacement figure that complies with the CC BY 4.0 license. Please check copyright information on all replacement figures and update the figure caption with source information. If applicable, please specify in the figure caption text when a figure is similar but not identical to the original image and is therefore for illustrative purposes only.

Reviewers' comments:

Reviewer's Responses to Questions

**Comments to the Author**

1. Is the manuscript technically sound, and do the data support the conclusions?

Reviewer #1: Yes

Reviewer #2: Partly

Reviewer #3: Yes

2. Has the statistical analysis been performed appropriately and rigorously? 

Reviewer #1: Yes

Reviewer #2: I Don't Know

Reviewer #3: No

3. Have the authors made all data underlying the findings in their manuscript fully available?

Reviewer #1: Yes

Reviewer #2: No

Reviewer #3: Yes

4. Is the manuscript presented in an intelligible fashion and written in standard English?

Reviewer #1: Yes

Reviewer #2: Yes

Reviewer #3: Yes

5. Review Comments to the Author

Reviewer #1: General comments

Anthropogenic disturbance impacts on biodiversity are of concern globally. Here the authors assessed roost disturbance of straw-coloured fruit bats (Eidolon helvum) through a novel technique of tri-axial acceleration. They have assumed that all flights during the daytime are a consequence of human disturbance. My only concern is how much bat flight occurs during the daytime at roosts because of interactions between bat individuals? Although you used daytime flights as a proxy for disturbance were you able to validate any of these? The manuscript is presented well.

Specific comments

Title:

Fine.

Abstract:

Generally fine.

L32 maybe use the common name instead of just ‘bats’.

L32 put GPS in full.

Introduction:

Good synthesis of the relevant literature.

L55 put GPS in full.

Methods:

Generally well presented.

L96 Does your University/ Institute not require you to have animal ethics permission/ clearance from them for your proposed study?

L133 delete hours.

L135 perhaps replace ‘deployed’ with ‘fitted’.

L137 insert type of glue in parentheses after ‘glue’.

L138 insert ‘neck’ before ‘collar’.

L141 delete hours.

L142 abbreviate ‘minutes’.

L143 data is a plural so change ‘was’ to ‘were'.

L151 delete hours.

L153 change to ‘compared with’.

L162 change to ‘analyses’.

L166 delete hours.

L178 reword to ‘of bats flying during the daytime’.

L179, 180 change to ‘analyses’.

L183 replace ‘propagate’ with ‘spread’.

L192 change to ‘analyses’.

L198 replace ‘due to’ with ‘because of’.

Results:

Generally fine.

L227 insert ‘a’ before ‘model’.

Discussion:

Good synthesis.

L267 insert Latin name.

References:

I have not checked if Journal format was followed.

Reviewer #2: Comments to the authors

I hereby provide comments for the manuscript Assessing roost disturbance of straw-coloured fruit bats (Eidolon helvum) through tri-axial acceleration.

The study that I reviewed has two goals: 1) test a new method for the assessment of disturbance to bat roosts and 2) identify causes of disturbance to the straw-coloured fruit bat. The manuscript is well written and structured, it is easy to follow and I think it constitutes a nice contribution. I have two main comments and a number of minor suggestions, the latter directed at improving clarity at specific points in the manuscript. I will highlight when they are just a suggestion and the change can be flexible.

Comment 1: I have some concerns regarding the structure of the models.

As you mention also in your discussion, there can be factor that affect flight activity linked to the individual colony that are not addressed. It would be maybe relevant to have ‘location’ as random factor to account for those, or at least to check this as you do for the individual level.

The other point about the models is for the use of ‘human density’. It is not clear to me if A) there are 4 unique values only (one for each location/point of capture, the black ones in Figure 1), or if B) the ‘human density’ variable is calculated around each identified roost (both black and white points in Figure 1). From one of the graphs I can think maybe option B) because the rugplot at the bottom of the graph seems to show multiple lines, but it should be made more clear. If instead it is case A), I think there might be a statistical problem – ‘location’ could be confounded with ‘human density’, and ‘human density’ might not qualify to be a continuous variable as I believe the effective sample size is 4 (correct if I’m wrong). In this case, ‘location’ could become a categorical fixed factor and conclusions about ‘human density’ could be made only at discussion level if the most ‘disturbed’ area is also the one with highest ‘human density’.

Comment 2: I think the manuscript puts too much emphasis on results from model 1 in both the discussion and, especially, abstract. Or better, should acknowledge that there are contrasting results. Yes, model 2 has a lower R^2, but both are not very high and I can’t think why removing the capture days from the dataset should reduce the impact of ‘human density’. You could also explore if conducting model selection would highlight some of the main effects.

In either cases, the manuscript could also list the testing of the method as a goal in the aim section of the introduction.

Minor comments:

Line 49. ‘species distributions’

Line 50. Suggestion: ‘…is a challenging task, as impacts vary across species and contexts.’

Line 54. the use of automated methods is has become more common – to answer the task described in the previous paragraph (assessing the impact of disturbance) or for studies in general?

Lines 55-57. They do reduce the influence of the observer, but we still know very little of how the behaviour in bats is affected by relatively heavy tags (>5% of the bat weight), although it is likely that they provide some sort of disturbance. Maybe you could revise this sentence to either mention the duality in their use or highlight different benefits connected to your study.

Line 58. ‘new sensors for’ -> ‘new sensors to measure’

Lines 69-70. worth mentioning a few more detail about the roosting habitat? (for example, if they have particular requirements in terms of forest density/type, etc.)

Line 71. Add Country name where park is located.

Line 72. ‘making thenm’

Line 77-78. A small step seems to be missing here regarding as to why daytime flights can be used a proxy for disturbance. Also ‘which where’ -> ‘and used them’

Line 83. ‘predicted’ -> ‘assumed’

Line 84. ‘the colonies that..’ -> ‘the colonies, and we predicted that this…’

Line 104. ‘Black’

Lines 108-112. Descriptions of the other study areas also briefly mention vegetation type, but here it is not described. Also, sometimes the word ‘areas’ is used and sometimes ‘locations’ (see for example line 98 and Table 1). Change for consistency.

Lines 113-117. Mention if the two sampling points in this area are regarded as one (for example, if environmental variables are considered homogeneous across the two)

Line 121. 70,000

Line 127. Suggestion: For structure consistency mention which season (wet or dry) October to December represent.

Line 133: ‘we weighted the captured bats’

Line 138. Mention glue type

Line 141. How often were GPS points recorded?

Line 154. ‘torsion’ -> ‘variation’. Is this change correct? If I understand correctly, heave, surge and sway are translational motions (the ones you can identify with the loggers), while rotational motions (torsion) would be pitch, roll and yaw.

Lines 156-157. Any reference for this method of assessing flying motion?

Lines 157-159. Please rephrase this sentence and give also a less technical explanation of why that ensures correct positioning (similar to the way you explain flight in lines 153-154).

Lines 167-168. To further improve understanding, use also the expression ‘occurrence of daytime roost flights’ here (to let the reader easily understand that you will use a binary variable).

Line 168-169. Why do you exclude bats that travel more than 500 meters? Explain please.

Line 170. Here you say you remove the first day of tracking of the animals, but isn’t this the ‘capture day’ that it is later on included in the first model version?

Line 178. ‘probably -> ‘probability’

Lines 176-193. As it can be inferred from my other comments, I think here there is some detail missing regarding the covariate used and the levels at which they are defined. Is every bat from a certain area/location going to have the same solar irradiance? Or are these variables defined with higher spatial resolution? What about human density? These details are important to evaluate the suitability of model structure. Also, for the effect of capture activity it should be said explicitly here or in the next paragraph that it is binary yes/no for each study unit (correct?).

Line 196. ‘bat captures’ -> ‘bat capture activity’

Line 200. So, your analysis unit is ‘occurrence of daytime roost flight’ defined for each bat and each day of tracking (rephrase as you wish, but I suggest to state it clearly somewhere in the text).

Line 212-214. You can also provide some basic information on how many days had flight occurrence or what are the ranges of the variables measured (for example, if human density is defined at the location density, you can mention values for each location, or similar info). This helps contextualise the results.

Line 241. Maybe write the actual number instead of the number of observations excluded.

Reviewer #3: This study assesses roost disturbance in straw-coloured fruit bats using GPS-accelerometers to measure the probability of daytime flights using the bat capture days as a proxy for human disturbance. Bat poaching is of increasing concern not only for bat conservation but also zoonotic diseases, so new methods to detect and deter poachers are of importance. I am not totally convinced by the efficacy of the method at the current time (e.g. short tracking duration, weight of devices and relatively labour intensive), but it is likely the technology will advance to a point where it becomes viable so methods making use of these results show promise for the future. The study uses bats from 5 different sites and has a good sample size from each. The only major comment I have is that the current analyses do not take into account the fact that there are five different sites and the temporal relatedness of variables (see below). I’m uncertain whether the changes to the model I have suggested below will alter the results, but it needs to be incorporated to check that the results for human density in particular are not just a result of other unmeasured differences between sites. I confident that you will be able to make these changes, and I congratulate you on an otherwise well-written and executed paper.

Major comments:

As far as I can tell, you have not accounted for the effect of colony location in your analysis. The study design is a nested repeated measures design where you have 5 colonies with several individuals tracked in each colony during different seasons and for varying numbers of days. Given the human density metric will be the same for each bat at the same colony (and to a lesser degree the weather variables as sampling occurred over numerous days), this means that your current model essentially makes the assumption that all of the differences between sites are due to human density and does not allow for any unmeasured differences between sites, which is unlikely given the large spatial and temporal distribution of sites. There are a couple of different ways to solve this problem, such as nesting random effects, but I think I would personally add the location as a categorical fixed-effect to measure if there are differences between the 5 sites (see below).

The second problem is that the temporal relatedness of the capture dates is not taken into account. Temporal autocorrelation predicts that observations taken closely together are more likely to be related to each other e.g. the bats captured on the same day are likely to be more related to each other (particularly in colonies) than the bats captured a week later. For example, bat 1607 was captured on the 4th February and tracked for 5 days, which overlaps with bat 1616 captured on the 6th February and bat 1620 captured on the 7th February. For your study, I recommend that you nest date within each site because the dates at different sites are unlikely to impact the results. For further reading on autocorrelation, see Pinheiro, J. C., & Bates, D. (2000). Mixed-effects models in S and S-PLUS; and Boyce et. al. (2010) Temporal autocorrelation functions for movement rates from global positioning system radiotelemetry data.

Last time I checked, adding temporal autocorrelation could not be done easily in lme4 but it can be done for a binomial model using a combination of nlme and MASS (or potentially a Bayesian package like brms). In nlme and MASS, your R code would look something like:

library(MASS)

library(nlme)

model <- glmmPQL(flights ~ capture + humans + solar + precipitation + wind + location, random = ~ 1 | individual, correlation=corAR1(form = ~ date| location), family = binomial, data = your_data)

(corAR1 = Autoregressive Lag-1 correlation).

This model will take some time to run as it will have to do repeated calls. Please do conduct your own research to check the model I have written here is correct as I have done this relatively quickly without your data, particularly the syntax on the autocorrelation as I have not nested that before. It goes without saying that you need to add back in your individual random effects. There is no reason to remove individual random effects on unbalanced designs.

Minor comments:

Line 55: Comma required after Ref 19 to create a clause – otherwise the tense is wrong.

68: assumed to decline – assumed to be declining? More specific would be better. Are there any studies quantifying this?

72: then – them

Methods: Please add the sample sizes into the main text either in the study area descriptions and/or Table 1.

Conclusions: The results of accelerometer studies can also be used to refine other methods for detecting poachers, such as motion capture methods. You could add a couple of sentences in the discussion about this.

6. PLOS authors have the option to publish the peer review history of their article (what does this mean?). If published, this will include your full peer review and any attached files.

Reviewer #1: No

Reviewer #2: No

Reviewer #3: **Yes: **Lucy A. Taylor

---

## [Author Response · Author response to Decision Letter 0]

30 Sep 2020

Editor Comments:

All the reviewers agree that the study is sound and interesting. Please address all the minor suggestions and also see particularly to the insightful comment by reviewers 2 and 3 on the statistical analysis.

Response: We agree that the suggestions of reviewers 2 and 3 on the statistical analysis were appropriated and made changes in our models in accordance. 

Copyright of map used in Figure 1.

Response: The background map of Figure 1 was developed by Database of Global Administrative Areas (GADM) and its use is allowed for academic publishing (https://gadm.org/license.html). This information was added to the caption of Figure 1.

Reviewer #1: General comments

Anthropogenic disturbance impacts on biodiversity are of concern globally. Here the authors assessed roost disturbance of straw-coloured fruit bats (Eidolon helvum) through a novel technique of tri-axial acceleration. They have assumed that all flights during the daytime are a consequence of human disturbance. My only concern is how much bat flight occurs during the daytime at roosts because of interactions between bat individuals? Although you used daytime flights as a proxy for disturbance were you able to validate any of these? The manuscript is presented well.

Response: We clarified in lines 82-89 that we investigated human disturbance (caused by our presence in the colonies during trapping) but also natural stressors described in the literature to influence the probability of bats to fly during the day. We think the removal of human density, as suggested by reviewers #2 and #3, helped to make a clear distinction between human and environmental stressors.

We agree that some flights in the colonies may have occurred due to interactions between individuals. However, we have no reason to think that those interactions will influence or invalidate the relationships that emerged from our models.

With regards to the use of daytime flights as a proxy for disturbance, we added a justification and a reference in lines 80-82. 

Specific comments

Title:

Fine.

Abstract:

Generally fine.

L32 maybe use the common name instead of just ‘bats’.

Response: Corrected.

L32 put GPS in full.

Response: Added.

Introduction:

Good synthesis of the relevant literature.

L55 put GPS in full.

Response: Added.

Methods:

Generally well presented.

L96 Does your University/ Institute not require you to have animal ethics permission/ clearance from them for your proposed study?

Response: The Max Planck Institute of Animal Behavior, the leading institution of this project, required only permits from authorities of the countries where fieldwork took place. Those permits accounted for national regulations for ethics in animal experimentation and the use wild animals in scientific research in each country. 

L133 delete hours.

Response: Deleted.

L135 perhaps replace ‘deployed’ with ‘fitted’.

Response: Changed.

L137 insert type of glue in parentheses after ‘glue’.

Response: Added.

L138 insert ‘neck’ before ‘collar’.

Response: Added.

L141 delete hours.

Response: Deleted.

L142 abbreviate ‘minutes’.

Response: Abbreviated.

L143 data is a plural so change ‘was’ to ‘were'.

Response: Corrected.

L151 delete hours.

Response: Deleted.

L153 change to ‘compared with’.

Response: Corrected.

L162 change to ‘analyses’.

Response: Corrected.

L166 delete hours.

Response: Deleted.

L178 reword to ‘of bats flying during the daytime’.

Response: Changed.

L179, 180 change to ‘analyses’.

Response: Corrected.

L183 replace ‘propagate’ with ‘spread’.

Response: Changed.

L192 change to ‘analyses’.

Response: This sentence was removed as a result of changes described in our response to comment 1 of Reviewer #2.

L198 replace ‘due to’ with ‘because of’.

Response: This sentence was removed as a result of changes described in our response to comment 1 of Reviewer #2.

Results:

Generally fine.

L227 insert ‘a’ before ‘model’.

Response: Corrected.

Discussion:

Good synthesis.

L267 insert Latin name.

Response: Added.

References:

I have not checked if Journal format was followed.

Reviewer #2: Comments to the authors

I hereby provide comments for the manuscript Assessing roost disturbance of straw-coloured fruit bats (Eidolon helvum) through tri-axial acceleration.

The study that I reviewed has two goals: 1) test a new method for the assessment of disturbance to bat roosts and 2) identify causes of disturbance to the straw-coloured fruit bat. The manuscript is well written and structured, it is easy to follow and I think it constitutes a nice contribution. I have two main comments and a number of minor suggestions, the latter directed at improving clarity at specific points in the manuscript. I will highlight when they are just a suggestion and the change can be flexible.

Comment 1: I have some concerns regarding the structure of the models.

As you mention also in your discussion, there can be factor that affect flight activity linked to the individual colony that are not addressed. It would be maybe relevant to have ‘location’ as random factor to account for those, or at least to check this as you do for the individual level.

The other point about the models is for the use of ‘human density’. It is not clear to me if A) there are 4 unique values only (one for each location/point of capture, the black ones in Figure 1), or if B) the ‘human density’ variable is calculated around each identified roost (both black and white points in Figure 1). From one of the graphs I can think maybe option B) because the rugplot at the bottom of the graph seems to show multiple lines, but it should be made more clear. If instead it is case A), I think there might be a statistical problem – ‘location’ could be confounded with ‘human density’, and ‘human density’ might not qualify to be a continuous variable as I believe the effective sample size is 4 (correct if I’m wrong). In this case, ‘location’ could become a categorical fixed factor and conclusions about ‘human density’ could be made only at discussion level if the most ‘disturbed’ area is also the one with highest ‘human density’. 

Response: We agree with the reviewer and this comment is consistent with that of reviewer #3 (major comments first paragraph). As mentioned by both reviewers, colonies may vary in several factors with potential influence on the flight behaviour of bats, with human density being among those factors. We followed the suggestions of the reviewers and re-fitted our models with colony ID and individual ID as random factors and excluding human density as predictor (Table 2 and Figure 3 were updated with the results of the new models, as well as the text in lines 204-215, 231-237, 241-244, 282-283). Compared with the models built before, the inclusion of colony ID slightly influenced the variation explained by individual ID. That is the reason why we decided to include individual ID as random factor in our models. The results are similar to the earlier models but the exclusion of human density required changes across the manuscript (see lines 35, 83-88, 201, 229, 269-270).

Comment 2: I think the manuscript puts too much emphasis on results from model 1 in both the discussion and, especially, abstract. Or better, should acknowledge that there are contrasting results. Yes, model 2 has a lower R^2, but both are not very high and I can’t think why removing the capture days from the dataset should reduce the impact of ‘human density’. You could also explore if conducting model selection would highlight some of the main effects.

Response: We think this issue is solved with the new models, for which there is no inconsistency in the significance of the common predictors. However, the new results reinforce the idea that the differences in R2 between models are mainly due to sample size (see Table 2). 

Regarding the model selection suggestion, we used an information-theoretic approach, rather than frequentist, as we did not have an exhaustive collection of potential predictors of flight behaviour. Following this approach, we drew a clear hypothesis for each variable used as predictor (lines 85-89), supported by the existing literature, and used the models to test those hypotheses. We think model selection would be more appropriated in a frequentist approach. 

In either cases, the manuscript could also list the testing of the method as a goal in the aim section of the introduction.

Response: Added in lines 76-77

Minor comments:

Line 49. ‘species distributions’

Response: Corrected.

Line 50. Suggestion: ‘…is a challenging task, as impacts vary across species and contexts.’

Response: Changed.

Line 54. the use of automated methods is has become more common – to answer the task described in the previous paragraph (assessing the impact of disturbance) or for studies in general?

Response: Clarified.

Lines 55-57. They do reduce the influence of the observer, but we still know very little of how the behaviour in bats is affected by relatively heavy tags (>5% of the bat weight), although it is likely that they provide some sort of disturbance. Maybe you could revise this sentence to either mention the duality in their use or highlight different benefits connected to your study.

Response: The sentence was revised and a reference of disturbance effects of biologging was added.

Line 58. ‘new sensors for’ -> ‘new sensors to measure’

Response: Corrected.

Lines 69-70. worth mentioning a few more detail about the roosting habitat? (for example, if they have particular requirements in terms of forest density/type, etc.)

Response: This species does not show particular preference for trees species or forest types as roosting habitat (see reference [13]), except that it often roosts in urban parks as specified in lines 72-74.

Line 71. Add Country name where park is located.

Response: Added.

Line 72. ‘making thenm’

Response: Corrected.

Line 77-78. A small step seems to be missing here regarding as to why daytime flights can be used a proxy for disturbance. 

Response: This was also pointed out by reviewer #1. We added a justification and a reference in lines 80-82.

Also ‘which where’ -> ‘and used them’

Response: Corrected.

Line 83. ‘predicted’ -> ‘assumed’

Response: This sentence was changed to address the issues raised in comment 1 of this reviewer.

Line 84. ‘the colonies that..’ -> ‘the colonies, and we predicted that this…’

Response: This sentence was changed to address the issues raised in comment 1 of this reviewer.

Line 104. ‘Black’

Response: Corrected.

Lines 108-112. Descriptions of the other study areas also briefly mention vegetation type, but here it is not described. Also, sometimes the word ‘areas’ is used and sometimes ‘locations’ (see for example line 98 and Table 1). Change for consistency.

Response: A brief description of vegetation was added as suggested (lines 110-111). References to “site” or “location” were changed to “area” for consistency (lines 100, 132, 150, 201 and Table 1).

Lines 113-117. Mention if the two sampling points in this area are regarded as one (for example, if environmental variables are considered homogeneous across the two)

Response: All roosts were considered separately in the analysis. We think this becomes clear from Fig 1. However, weather variables were retrieved from the same stations in roosts close together. This was clarified in lines 197-199.

We also added the distance between the two main colonies in Kibi (line 114). 

Line 121. 70,000

Response: Corrected.

Line 127. Suggestion: For structure consistency mention which season (wet or dry) October to December represent.

Response: Added.

Line 133: ‘we weighted the captured bats’

Response: Corrected.

Line 138. Mention glue type

Response: Added.

Line 141. How often were GPS points recorded?

Response: Added.

Line 154. ‘torsion’ -> ‘variation’. Is this change correct? If I understand correctly, heave, surge and sway are translational motions (the ones you can identify with the loggers), while rotational motions (torsion) would be pitch, roll and yaw.

Response: Corrected.

Lines 156-157. Any reference for this method of assessing flying motion?

Response: Using accelerometers to identify flight of animals is a relatively common method. For that we cited a review paper in line 156. However, the calculations described in lines 157-168 are specific for this dataset. 

Lines 157-159. Please rephrase this sentence and give also a less technical explanation of why that ensures correct positioning (similar to the way you explain flight in lines 153-154).

Response: A more complete explanation is now provided in lines 163-165. Further changes were made in lines 157-162 in order to clarify the classification criteria used.

Lines 167-168. To further improve understanding, use also the expression ‘occurrence of daytime roost flights’ here (to let the reader easily understand that you will use a binary variable).

Response: A sentence as added in lines 174-176 to clarify this aspect. 

Line 168-169. Why do you exclude bats that travel more than 500 meters? Explain please.

Response: Explanation added (lines 177-178).

Line 170. Here you say you remove the first day of tracking of the animals, but isn’t this the ‘capture day’ that it is later on included in the first model version?

Response: “Capture day” used across the manuscript refer to the days that we visited the colony for captures in general. Our presence was assumed to disturb the whole colony, as described in lines 186-192. We also assumed that all bats with dataloggers deployed earlier would reflect the disturbance of the colony (added in line 192). Apart from that, we expect the capture and handling of each bat would affect its behaviour for the rest of the day, thus we decided to exclude the data from the first tracking day of each bat.

We added this explanation in lines 178-179 and we also re-worded several parts of the manuscript for clarification (lines 34-35, 37, 186, 224, 225, 226, 234, 235, 241, 247, 261, 263, Table 1 model and variable names). 

Line 178. ‘probably -> ‘probability’

Response: Corrected.

Lines 176-193. As it can be inferred from my other comments, I think here there is some detail missing regarding the covariate used and the levels at which they are defined. Is every bat from a certain area/location going to have the same solar irradiance? Or are these variables defined with higher spatial resolution? What about human density? These details are important to evaluate the suitability of model structure. 

Response: We now explain in lines 197-199 that weather variables were retrieved from the same stations in roosts placed close together. This doesn’t mean that there is much repetition of values in our dataset as there is little time overlap between the different bats tracked (see Table S1) and weather data were updated every day. Human density was excluded from the analysis, as detailed above in our response to comment 1 of this reviewer. 

Also, for the effect of capture activity it should be said explicitly here or in the next paragraph that it is binary yes/no for each study unit (correct?).

Response: Added in lines 210-211 and also in Table 2

Line 196. ‘bat captures’ -> ‘bat capture activity’

Response: We edited this sentence to address the issues described above (reviewer comment to line 70)

Line 200. So, your analysis unit is ‘occurrence of daytime roost flight’ defined for each bat and each day of tracking (rephrase as you wish, but I suggest to state it clearly somewhere in the text).

Response: We now state this clearly in lines 174-176.

Line 212-214. You can also provide some basic information on how many days had flight occurrence or what are the ranges of the variables measured (for example, if human density is defined at the location density, you can mention values for each location, or similar info). This helps contextualise the results.

Response: The number of daytime roost flight occurrences was added to the text (line 223). The range of the variables used as model predictors was included in Table 2.

Line 241. Maybe write the actual number instead of the number of observations excluded.

Response: Added

Reviewer #3: Comments to the authors

This study assesses roost disturbance in straw-coloured fruit bats using GPS-accelerometers to measure the probability of daytime flights using the bat capture days as a proxy for human disturbance. Bat poaching is of increasing concern not only for bat conservation but also zoonotic diseases, so new methods to detect and deter poachers are of importance. I am not totally convinced by the efficacy of the method at the current time (e.g. short tracking duration, weight of devices and relatively labour intensive), but it is likely the technology will advance to a point where it becomes viable so methods making use of these results show promise for the future. The study uses bats from 5 different sites and has a good sample size from each. The only major comment I have is that the current analyses do not take into account the fact that there are five different sites and the temporal relatedness of variables (see below). I’m uncertain whether the changes to the model I have suggested below will alter the results, but it needs to be incorporated to check that the results for human density in particular are not just a result of other unmeasured differences between sites. I confident that you will be able to make these changes, and I congratulate you on an otherwise well-written and executed paper.

Response: We agree that the method described will be more valuable in the near future than when the data was collected (2009-2014). Actually, if the data collection was done in the current days, we would have available similar dataloggers at much lower costs, with longer lifespan and the remote transmission of data, which would reduce the costs and fieldwork effort considerably. This idea was already in lines 302-305 but we added a complement in lines 306-307. 

We describe below how we addressed the remaining issues.

Major comments:

As far as I can tell, you have not accounted for the effect of colony location in your analysis. The study design is a nested repeated measures design where you have 5 colonies with several individuals tracked in each colony during different seasons and for varying numbers of days. Given the human density metric will be the same for each bat at the same colony (and to a lesser degree the weather variables as sampling occurred over numerous days), this means that your current model essentially makes the assumption that all of the differences between sites are due to human density and does not allow for any unmeasured differences between sites, which is unlikely given the large spatial and temporal distribution of sites. There are a couple of different ways to solve this problem, such as nesting random effects, but I think I would personally add the location as a categorical fixed-effect to measure if there are differences between the 5 sites (see below).

Response: We agree with the reviewer. This issue was also pointed by reviewer #2 (major comment 1). We followed the suggestions of both reviewers and re-fitted our models with colony ID and individual ID as random factors and excluding human density as a predictor (Table 2 and Figure 3 were updated with the results of the new models, as well as the text in lines 204-215, 231-237, 241-244, 282-283). The inclusion of colony ID slightly influenced the variation explained by individual ID, therefore we decided to include individual ID as random factor in our models. The results are similar to the earlier models but the exclusion of human density required changes across the manuscript (see lines 35, 83-88, 201, 229, 269-270).

We should clarify that the roost ID included 30 different roosts (not 5, see Fig 1). 

The second problem is that the temporal relatedness of the capture dates is not taken into account. Temporal autocorrelation predicts that observations taken closely together are more likely to be related to each other e.g. the bats captured on the same day are likely to be more related to each other (particularly in colonies) than the bats captured a week later. For example, bat 1607 was captured on the 4th February and tracked for 5 days, which overlaps with bat 1616 captured on the 6th February and bat 1620 captured on the 7th February. For your study, I recommend that you nest date within each site because the dates at different sites are unlikely to impact the results. For further reading on autocorrelation, see Pinheiro, J. C., & Bates, D. (2000). Mixed-effects models in S and S-PLUS; and Boyce et. al. (2010) Temporal autocorrelation functions for movement rates from global positioning system radiotelemetry data.

Last time I checked, adding temporal autocorrelation could not be done easily in lme4 but it can be done for a binomial model using a combination of nlme and MASS (or potentially a Bayesian package like brms). In nlme and MASS, your R code would look something like:

library(MASS)

library(nlme)

model <- glmmPQL(flights ~ capture + humans + solar + precipitation + wind + location, random = ~ 1 | individual, correlation=corAR1(form = ~ date| location), family = binomial, data = your_data)

(corAR1 = Autoregressive Lag-1 correlation).

This model will take some time to run as it will have to do repeated calls. Please do conduct your own research to check the model I have written here is correct as I have done this relatively quickly without your data, particularly the syntax on the autocorrelation as I have not nested that before. It goes without saying that you need to add back in your individual random effects. There is no reason to remove individual random effects on unbalanced designs.

Response: We were aware of the possible temporal autocorrelation issues mentioned, as they are common in tracking studies, particularly those using high temporal resolution data. However, the aggregation of data into daily values (described in lines 173-176) was likely to reduce or cancel temporal autocorrelation of model residuals.

We now conducted a formal analysis of temporal autocorrelation of our model residuals and present the results in Fig S1 (with an explanation added in lines 214-215). As shown in Fig S1, correlation of model residuals does not show a temporal trend, thus we think that adding a temporal correlation structure to our models would lead to unnecessary complexity. But we are very grateful for the detailed explanation, literature and R code with possible solutions to account for temporal autocorrelation.

We should emphasize that there was no time overlap between the data collected in the four different regions (shown in Tables 1 and S1). Thus, we were able to use the full dataset in our temporal autocorrelation analysis. 

Minor comments:

Line 55: Comma required after Ref 19 to create a clause – otherwise the tense is wrong.

Response: Corrected.

68: assumed to decline – assumed to be declining? More specific would be better. Are there any studies quantifying this?

Response: “assumed to” was removed. The conclusions of UICN most recent evaluation (reference [27]) states that this species in indeed declining.

72: then – them

Response: Corrected.

Methods: Please add the sample sizes into the main text either in the study area descriptions and/or Table 1.

Response: Added in Table 1.

Conclusions: The results of accelerometer studies can also be used to refine other methods for detecting poachers, such as motion capture methods. You could add a couple of sentences in the discussion about this.

Response: Added in lines 300-302 and a supporting reference was also included.

---

## [Decision Letter · Decision Letter 1]

5 Nov 2020

PONE-D-20-15313R1

Assessing roost disturbance of straw-coloured fruit bats (Eidolon helvum) through tri-axial acceleration

PLOS ONE

Dear Dr. Santos,

Thank you for submitting your manuscript to PLOS ONE. After careful consideration, we feel that it has merit but does not fully meet PLOS ONE’s publication criteria as it currently stands. Therefore, we invite you to submit a revised version of the manuscript that addresses the points raised during the review process.

We look forward to receiving your revised manuscript.

Kind regards,

Christian Vincenot, Ph.D.

Academic Editor

PLOS ONE

Additional Editor Comments (if provided):

The manuscript is basically fit for publication as far as I am concerned. Please just address the few remaining minor comments by reviewer 2 (esp. the supplementary data requested) and resubmit, after which I will promptly send the acceptance letter. Congratulations on the interesting study.

Reviewers' comments:

Reviewer's Responses to Questions

**Comments to the Author**

1. If the authors have adequately addressed your comments raised in a previous round of review and you feel that this manuscript is now acceptable for publication, you may indicate that here to bypass the “Comments to the Author” section, enter your conflict of interest statement in the “Confidential to Editor” section, and submit your "Accept" recommendation.

Reviewer #1: All comments have been addressed

Reviewer #3: All comments have been addressed

2. Is the manuscript technically sound, and do the data support the conclusions?

Reviewer #1: Yes

Reviewer #3: Yes

3. Has the statistical analysis been performed appropriately and rigorously? 

Reviewer #1: Yes

Reviewer #3: Yes

4. Have the authors made all data underlying the findings in their manuscript fully available?

Reviewer #1: Yes

Reviewer #3: Yes

5. Is the manuscript presented in an intelligible fashion and written in standard English?

Reviewer #1: Yes

Reviewer #3: Yes

6. Review Comments to the Author

Reviewer #1: (No Response)

Reviewer #3: Thank you for the revised version of your manuscript (previously Reviewer 3). As before, this study assesses roost disturbance in straw-coloured fruit bats using GPS-accelerometers to

measure the probability of daytime flights using the bat capture days as a proxy for human

disturbance. The statistics are much improved without the use of the human density metric which was the same at each colony. There are still a few question marks I have over the analysis, particularly in relation to roost ID (see below), but I do not think these will have much impact on the results. I congratulate the authors on a well-written manuscript.

Minor comments:

The authors have used roost ID rather than colony as a random effect, which means for some colonies there are multiple roosts and others just the main colony site. For example, Ouagadougou seems to have 10 roosts whereas has Accra only has the main colony site. Although designed to reduce the bias of repeated measures, this design could give additional weight to certain colonies, particularly when you consider some of the weather data for each roost is taken from the same weather station (e.g. bats at different roosts on the same day would have the same solar irradiance value). I think any impacts would be very minor, and I also think you need to take the bat biology into account (e.g. significance of different roosts), which I am not able to do as I’m not a bat expert. It would be helpful if you could run your analyses using colony location as well and/or justify your decision to use roost in the manuscript when you refer to colony in the rest of the manuscript.

L55: has = have

Table 1: Please distinguish the two Kibi colonies as you refer to 5 colonies throughout the rest of the manuscript. Given that the two Kibi colonies are so close together and that the bats do seem to move roosts further than the distance between these two colonies, I’m not totally sure whether you should be referring to 4 or 5 colonies (when considering the spatial scale between these two sites and the other 3), but I’m not a bat expert so I leave this to you.

L208: If you use Roost ID as a random effect, please add an additional supplementary table detailing the sample size at each roost (bats and days). I presume this cannot be added to Table S1 as some of the bats will have swapped roost, but it’s helpful to see the overall distribution of your sample size.

7. PLOS authors have the option to publish the peer review history of their article (what does this mean?). If published, this will include your full peer review and any attached files.

Reviewer #1: No

Reviewer #3: No

---

## [Author Response · Author response to Decision Letter 1]

6 Nov 2020

Reviewer #3 comments

Thank you for the revised version of your manuscript (previously Reviewer 3). As before, this study assesses roost disturbance in straw-coloured fruit bats using GPS-accelerometers to measure the probability of daytime flights using the bat capture days as a proxy for human disturbance. The statistics are much improved without the use of the human density metric which was the same at each colony. There are still a few question marks I have over the analysis, particularly in relation to roost ID (see below), but I do not think these will have much impact on the results. I congratulate the authors on a well-written manuscript.

Minor comments:

The authors have used roost ID rather than colony as a random effect, which means for some colonies there are multiple roosts and others just the main colony site. For example, Ouagadougou seems to have 10 roosts whereas has Accra only has the main colony site. Although designed to reduce the bias of repeated measures, this design could give additional weight to certain colonies, particularly when you consider some of the weather data for each roost is taken from the same weather station (e.g. bats at different roosts on the same day would have the same solar irradiance value). I think any impacts would be very minor, and I also think you need to take the bat biology into account (e.g. significance of different roosts), which I am not able to do as I’m not a bat expert. It would be helpful if you could run your analyses using colony location as well and/or justify your decision to use roost in the manuscript when you refer to colony in the rest of the manuscript.

Response: We think we should account for possible roost specific effects on bat behaviour, which would not be accounted if we use colony site as random effect instead. Please note in Fig. 1 that roosts can be as far as 90 km from each other, thus potentially under different environmental conditions. Colony sites, on the other hand, do not have as much ecological meaning as they simply mark the places where the captures were conducted.

We anyway ran the analysis with colony site as random effect, as suggested, and the results were nearly identical (see the table below; the legend is the same as for Table 2 in the manuscript).

Model Parameter Range Estimate SE Z P-value R2 cond./marg

With trapping days Intercept - -7.169 2.033 -3.53 >0.001 0.24/0.22

 Trapping day 0-1 1.666 0.530 3.14 0.002 

 Solar irradiance 2.5-30.7 0.242 0.090 2.68 0.007 

 Precipitation 0.0-87.0 0.042 0.043 0.99 0.329 

 Wind speed 0.5-4.0 -0.320 0.377 -0.85 0.398 

Without trapping days Intercept - -6.008 1.831 -3.28 0.001 0.06/0.06

 Solar irradiance 2.5-30.7 0.175 0.089 1.96 0.050 

 Precipitation 0.0-87.0 0.034 0.042 0.82 0.410 

 Wind speed 0.5-4.0 -0.144 0.429 -0.34 0.738 

We agree that we used the term “colony” several times when we should have used “roost”. That was corrected across the manuscript. 

L55: has = have

Response: We think “has” should be used here as it refers to “The use of automated methods”. 

Table 1: Please distinguish the two Kibi colonies as you refer to 5 colonies throughout the rest of the manuscript. Given that the two Kibi colonies are so close together and that the bats do seem to move roosts further than the distance between these two colonies, I’m not totally sure whether you should be referring to 4 or 5 colonies (when considering the spatial scale between these two sites and the other 3), but I’m not a bat expert so I leave this to you.

Response: We made the changes suggested in Table 1. We now refer to roosts instead of colonies (see above), thus we maintained the reference to 5, instead of 4. 

L208: If you use Roost ID as a random effect, please add an additional supplementary table detailing the sample size at each roost (bats and days). I presume this cannot be added to Table S1 as some of the bats will have swapped roost, but it’s helpful to see the overall distribution of your sample size.

Response: We added a supplementary table (Table S2) containing the roost location for each bat in each day of tracking.

---

## [Editor Report · Decision Letter 2]

9 Nov 2020

Assessing roost disturbance of straw-coloured fruit bats (Eidolon helvum) through tri-axial acceleration

PONE-D-20-15313R2

Dear Dr. Santos,

We’re pleased to inform you that your manuscript has been judged scientifically suitable for publication and will be formally accepted for publication once it meets all outstanding technical requirements.

Kind regards,

Christian Vincenot, Ph.D.

Academic Editor

PLOS ONE

Additional Editor Comments (optional):

Thank you for addressing promptly the remaining comments. The manuscript is now accepted. Congratulations again on the interesting work!
---

## [Editor Report · Acceptance letter]

12 Nov 2020

PONE-D-20-15313R2 

Assessing roost disturbance of straw-coloured fruit bats (*Eidolon helvum*) through tri-axial acceleration 

Dear Dr. Santos:

I'm pleased to inform you that your manuscript has been deemed suitable for publication in PLOS ONE. Congratulations! Your manuscript is now with our production department. 

Kind regards, 

on behalf of

Dr. Christian Vincenot 

Academic Editor

PLOS ONE